# Tough double-bouligand architected concrete enabled by robotic additive manufacturing

Arjun Prihar[1], Shashank Gupta[1], Hadi S. Esmaeeli[1] & Reza Moini [1] ✉

Nature has developed numerous design motifs by arranging modest materials into complex architectures. The damage-tolerant, double-bouligand architecture found in the coelacanth fish scale is comprised of collagen fibrils helically arranged in a bilayer manner. Here, we exploit the toughening mechanisms of double-bouligand designs by engineering architected concrete using a large-scale two-component robotic additive manufacturing process. The process enables intricate fabrication of the architected concrete components at large-scale. The double-bouligand designs are benchmarked against bouligand and conventional rectilinear counterparts and monolithic casts. In contrast to cast concrete, double-bouligand design demonstrates a non-brittle response and a rising R-curve, due to a hypothesized bilayer crack shielding mechanism. In addition, interlocking behind and crack deflection ahead of the crack tip in bilayer double-bouligand architected concrete elicits a 63% increase in fracture toughness compared to cast counterparts.

In this work, we demonstrate a bio-inspired approach to the design of crack-resistant double-helical architected concrete materials through the use of advanced additive manufacturing processes and the exploitation of layered heterogeneity. Natural materials provide abundant examples of often mutually exclusive mechanical properties such as strength and fracture toughness by exploiting modest materials and weak interfaces arranged in purposeful architectures[1-8]. Compared to the design motifs of natural materials[9-11], construction materials such as masonry, stone, ceramics, and concrete are typically monolithic and suffer from a brittle failure response under tension[12]. Concrete, in particular, is the most commonly used human-made material, and yet it exhibits low fracture toughness and tensile strength[13], thus severely limiting its application.

Here, we propose engineering the toughening mechanisms found in the double-helical (double-bouligand) architecture, compared with helical (bouligand) architecture, in the design of concrete materials to overcome the often-mutual exclusivity between fracture toughness and tensile strength[14-17]. In particular, the double-helical architected concrete is proposed and benchmarked against helical architected concrete, rectilinear architected, and monolithic cast counterparts. We exploit the fabrication freedom and resolutions offered by a robotic

additive manufacturing technique that is thus far underutilized in order to achieve intricate architected concrete with advanced performance characteristics. In contrast to the rectilinear design approach commonly seen in additively manufactured concrete structures thus far[18-24], an architected design approach can stimulate mechanically advantageous mechanical responses that are otherwise unattainable using conventionally cast counterparts.

Though previous works take advantage of bouligand architectures in cementitious materials[15,17,25-27] to increase flexural strength[26], impact resistance (energy)[28], and work of failure (energy)[29], there has been no attempt to explicitly enhance fracture toughness (resistance to fracture) in bouligand architected materials. Here, the toughening mechanisms triggered by double-bouligand and bouligand[30] architectures, specifically crack shielding and crack twisting, are investigated in large-scale additively manufactured concrete materials. Particularly, the toughening mechanisms offered by the proposed double-helical structures were studied in comparison to helical and conventional rectilinearly manufactured, and monolithic cast counterparts.

The majority of concrete robotic additive manufacturing research focuses on the extrusion process and the material formulation to overcome the limits in build height[31] or overhang angle[32] of printed,

[1]Department of Civil and Environmental Engineering, Princeton University, Princeton, NJ, USA. ✉e-mail: Reza.moini@princeton.edu

self-supporting structures[33,34]. Moreover, the designs of additively manufactured materials have primarily relied on rectilinear or lamellar designs of materials[24,34,35], and consequently underutilizing the fabrication freedom offered by this technology. More specifically, thus far, no research has been conducted that harnesses concrete additive manufacturing technology to improve fracture toughness. In this work, we hypothesize that engineering the spatial arrangement of the concrete in a double-bouligand architecture can enable favorable mechanical properties that are otherwise only achieved through long-attempted efforts in optimizing the concrete material composition, such as through the addition of fibers in cast or additively manufactured counterparts (e.g., steel, PVA)[26,36,37].

With the advent of modern processing technologies (e.g., robotic additive manufacturing), the (meso) architecture of the material can be directly defined and, in turn, determine the material's properties and performances as illustrated in Fig. 1a. The structure of cement-based materials can be viewed by its microstructure as well as its (meso) architecture. The microstructure is the morphology and distribution of various phases[38] and the architecture can be defined as the purposeful arrangement of the bulk meso-scale materials to trigger specific mechanical responses[1]. This gives rise to architected cementitious materials in which the internal arrangement of the material is purposefully defined, designed, and controlled through a layer-wise additive scalable manufacturing process[15] that engenders improved mechanical properties (Fig. 1b).

The research on layer-wised additively manufactured concrete mainly focuses on mitigating the presence of weak interfaces that form between printed layers and result in anisotropic material properties[39–42]. In contrast, here we focus on exploiting weak interfaces to harness toughening mechanisms inspired by the double-helical architecture of the coelacanth fish scale (Fig. 2a–d) as compared with the helical architecture of the mantis shrimp dactyl club (Fig. 2e–h). The microstructure of these biological materials comprises modest constituent materials, such as collagen and chitin, yet are arranged in a manner to stimulate crack deflection and crack twisting toughening responses under load[30,43].

The double-bouligand architecture found in the scales of the coelacanth fish (Fig. 2a–c) is comprised of collagen fibril bundles arranged in parallel sheets[43,44]. However, each sheet is coupled with a second layer of fibrils oriented perpendicular to the sheet below it to form an orthogonal, bilayer unit[43]. This unit is then repeated in a helical pattern (Fig. 2d). The pitch angle, $\gamma$, in the double-bouligand is then defined by the relative rotation between each bilayer unit. As a result,

cracks alternate between deflection along a fibril interface and advancement through a layer of fibrils oriented perpendicular to the fracture plane[43]. This arrangement encourages crack deflection to prevent a brittle failure[45]. In the design of additively manufactured concrete in this study, the concrete filaments and porous interface emulate the collagen fibril bundle arrangement and the protein matrix, respectively.

The bouligand architecture found in the endocuticle of the mantis shrimp (Fig. 2e) dactyl club (Fig. 2f, g) is similar to the double-bouligand architecture, in that it is comprised of chitin fibrils arranged in parallel sheets joined by a protein matrix[11,30,46]. These sheets are stacked atop each other in a helical pattern defined by a pitch angle, $\gamma$, as the relative rotation between two sheets (Fig. 2h). The primary toughening mechanism is the twisting of impinging cracks preferentially through the weak, protein matrix interface between chitin fibrils. This twisting increases the fractured surface area, thereby increasing energy dissipation[30].

Robotic additive manufacturing offers control over the material architecture and, subsequently, the mechanical properties that are otherwise challenging to achieve with the current conventional casting methods used in construction. The recent growth of additive technologies[47] has led to a proliferation of single-material (commonly classified as 1-component or 1-K) gantry-based or robotic-based concrete additive manufacturing processes[45,48] (Supplementary Note 1, Supplementary Fig. 1). In contrast, a 2-component (2-K) extrusion system[49] (Fig. 3) exploited here, can facilitate more precise control of materials composition and mechanical properties through the secondary mixing of a set-accelerant at the nozzle (Supplementary Note 2). The early studies on the 2-K process merely examine mechanical properties such as compressive strength, explore mold-free applications with fiber-reinforced concrete, or study the processing and rheological aspects of material extrusion and deposition[50–53]. However, the 2-K robotic system has not yet been utilized to fabricate the purposeful and intricate design of material architectures.

We propose the use of a 2-K platform with advanced sensing and monitoring capabilities that can help engineer complex, structural-scale architected materials studied in this work (Fig. 3). We highlight the utility of the 2-K process to enhance the geometric fidelity of relatively complex design motifs at scale including columns (Fig. 3j), non-planer shells, and helical forms (Supplementary Fig. 2, Supplementary Videos 1–10).

Furthermore, aside from the optimization of geometries for load-bearing capacity[24,49,54], limited research has presented a digital

(a)

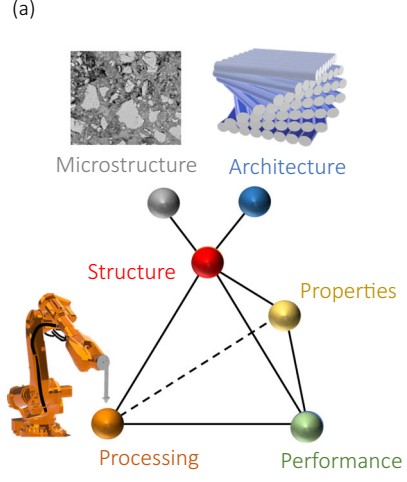

(b)

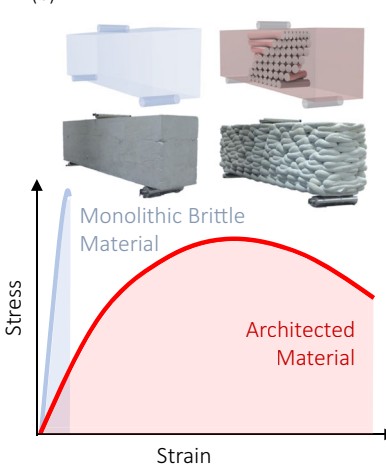

**Fig. 1 | Engineering material properties through the design of meso-scale architecture. a** Conceptual material science tetrahedron representing the interrelationships between the material's processing, structure (microstructure and meso architecture), properties, and performance, and **b** Schematic of the enhanced mechanical response of architected materials conceptualized compared to monolithic counterparts.

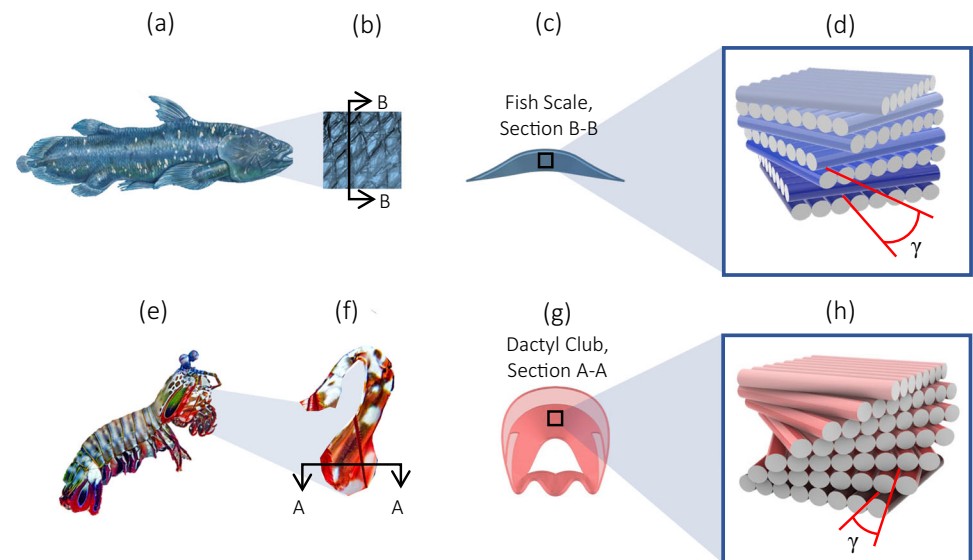

**Fig. 2 | Two bio-inspired motifs used in development of architected concrete. a** Image of the *Latimeria Chalumnae*, commonly known as the Coelacanth fish (By courtesy of Encyclopædia Britannica, Inc., copyright 2021; used with permission.), **b** Coelacanth scales, **c** Schematic cross-sectional view of fish scale, **d** Schematic representation of the collagen fibril bundles comprising the fish scales with a pitch angle, γ (inter-fibril arrangement not shown for clarity); **e** Image of the *Odontodactylus Scyllarus*, commonly known as the mantis shrimp (Credit: Adobe Stock: stock.adobe.com), **f** Dactyl club of mantis shrimp (Credit: Adobe Stock: stock.adobe.com), **g** Schematic cross-sectional view of dactyl club, **h** Schematic representation of chitin fibril arrangement in the endocuticle of the dactyl club with a pitch angle, γ.

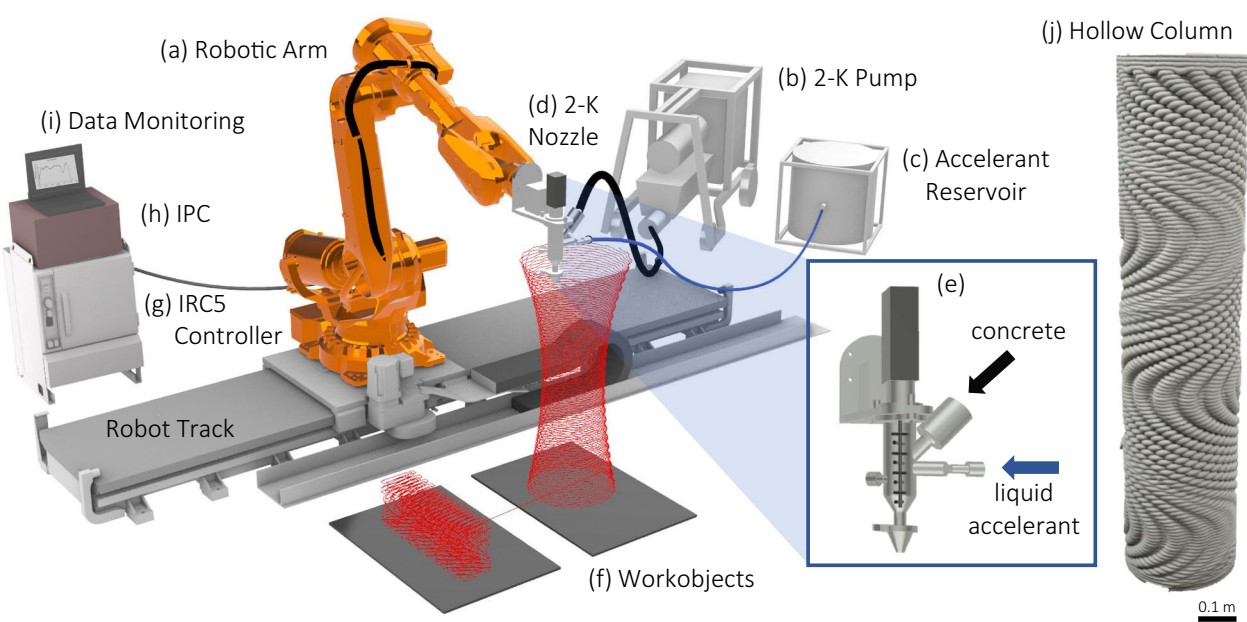

**Fig. 3 | Schematic of two-component (2-K) robotic additive manufacturing process. a** ABB IRB 6700 (2.85 m reach, 150 kg payload) positioned on a 5.7-meter track, **b** 2-K concrete pump with in-situ charging of feedstock, **c** Accelerant reservoir and dosing system, **d** 2-K agitation chamber featuring concrete and accelerant inlets, **e** Nozzle end-effector, **f** Workobject with visualized toolpath, **g** ABB IRC5 controller, **h** IPC system enabling digital control of concrete and accelerant pump flow rates, as well as nozzle chamber servo motor, **i** Monitoring real-time temperature and pressure sensor read-outs, **j** design opportunities to fabricate complex materials and structures such as an architected beam and a hollow column (that can act as lost formwork) using the 2-K process.

workflow for developing toolpaths to fabricate architected materials. A robust toolpath algorithm is necessary in order to facilitate the adoption of architected designs. The development approach for the toolpath algorithm developed in this study is further detailed in Supplementary Note 5.

In this work, we examine the toughening mechanisms of double-bouligand designs of architected concrete fabricated using a two-component robotic additive manufacturing process. In contrast to cast or additively manufactured rectilinear concrete, the double-bouligand design demonstrates enhanced fracture toughness and a rising R-curve. We hypothesize that bilayer crack shielding, in addition to interlocking behind and crack deflection ahead of the crack tip, in double-bouligand architected concrete elicits a significant increase in fracture toughness compared to cast counterparts.

## Results and discussion

The mechanical response and fracture properties of the proposed architected concrete, fabricated with double-bouligand and bouligand architectures, were characterized and compared to lamellar architectures and conventionally cast counterparts. The flexural strength and fracture toughness were characterized using unnotched and notched (Supplementary Fig. 7) specimens, respectively.

With regards to the flexural strength, along with a distinct difference observed in specimen stiffness (Fig. 4a), a statistically different modulus of rupture (MOR) was also recorded across the architected

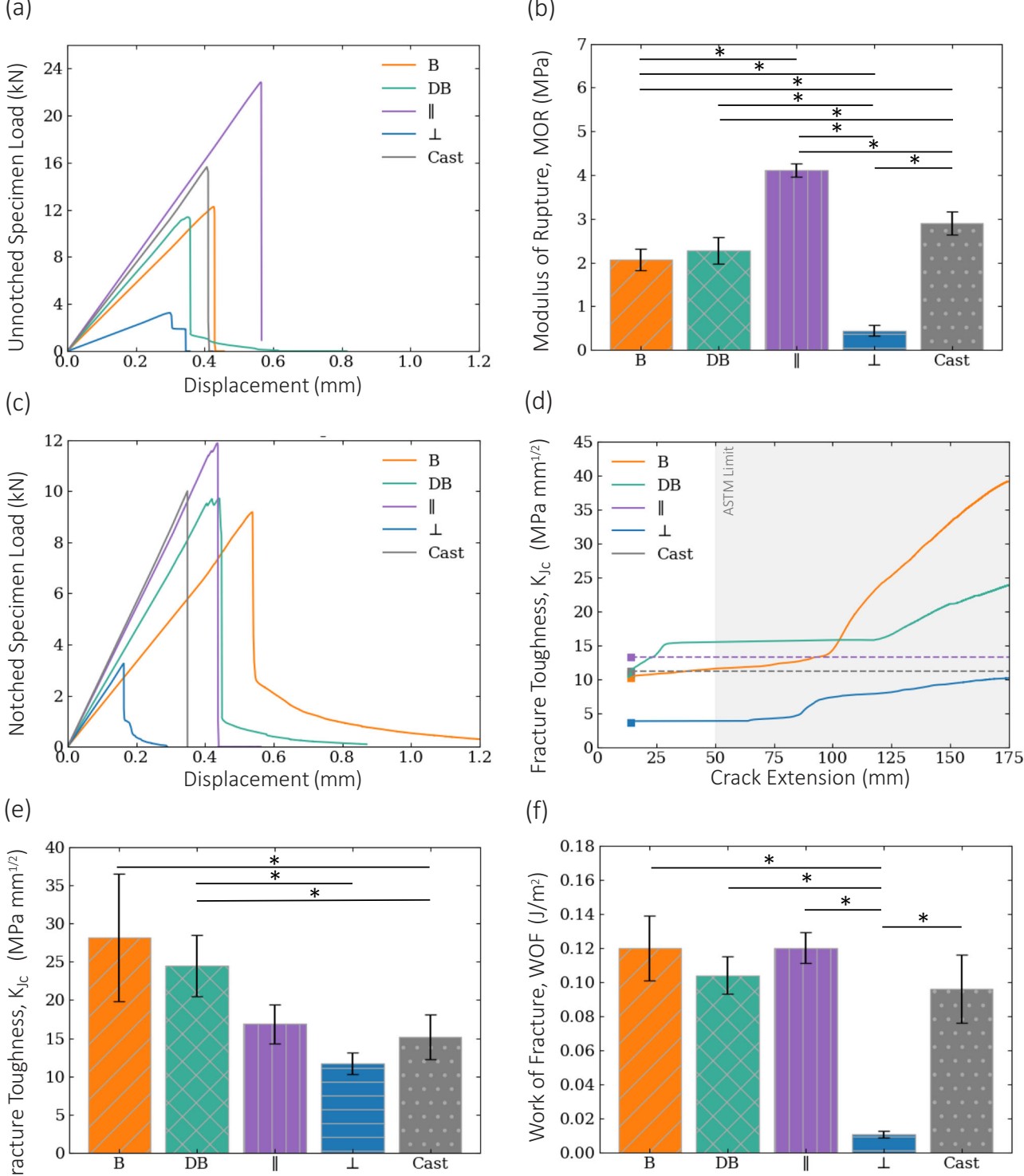

**Fig. 4 | Mechanical response and fracture properties of four designs of material architectures compared to monolithic cast: parallel lamellar, perpendicular lamellar, bouligand, and double-bouligand. a**, **b** Load-displacement plot and average modulus of rupture of unnotched specimens, **c** Load-displacement plot of notched specimens, **d** R-Curves (fracture toughness vs. crack extension) of notched specimens, with dashed lines denoting the highest fracture toughness for all design cases, and **e**, **f** Average toughness, $K_{Jc}$, and average work-of-fracture (WOF) of notched specimens. Data is shown as mean ± SD. * depicts $p < 0.05$ which indicates the statistically significant difference between the samples (at the ends of the solid line below a *). $p$-value is obtained from F-test and T-test.

and cast specimens (Fig. 4b). Evidently, the architected concrete materials exhibit statistically lower flexural strength than the cast counterparts of the same composition. It must be noted that the perpendicular lamellar architecture, with the interfaces oriented across the width of the specimen, had the lowest stiffness and statistically lowest flexural strength compared to all other specimens, as shown in Fig. 4a. Conversely, the parallel lamellar architecture, with interfaces oriented across the length, demonstrated the highest stiffness and statistically highest flexural strength compared to cast and architected materials. The stiffness of the double-bouligand and bouligand architected specimens resides between the two lamellar cases. Thus, we hypothesized that the stiffness of the architected material was dependent on the orientation of the interfaces. Most importantly, the double-bouligand architecture demonstrated a slight softening at the end of the load drop. We hypothesize that as the crack propagates through interfaces, it forms a twisted fracture plane governed by the pitch angle of the architecture. This twisted fracture plane is postulated to have generated interlocking between the two fractured surfaces of the specimen, thus resulting in additional load-bearing capacity past the peak load.

### Fracture response

The load-displacement plots of the notched specimens are shown in Fig. 4c. The notched specimens of the parallel lamellar, double-bouligand, bouligand, and cast reference all had statistically similar peak loads. While the cast and the parallel lamellar architecture demonstrated a brittle response, the architected, double-bouligand and

bouligand specimens remarkably presented a distinct post-peak softening response at the tail of the load-displacement response. The softening observed in the double-bouligand specimens occurred at $11.7 \pm 6.7$ % of the peak load, and the softening observed in the bouligand specimens occurred at $18.9 \pm 9.8$ % of the peak load.

This softening response in double-bouligand and bouligand specimens is indicative of toughening mechanisms that act to increase the resistance to fracture (i.e., fracture toughness). Using the relationship between crack extension and compliance presented in Eq. (9), R-curves are computed to present fracture resistance as a function of crack length (Fig. 4d). An increasing resistance to crack growth is commonly associated with extrinsic toughening mechanisms occurring behind the crack[3] that act to reduce the stress intensity at the crack tip[55].

Both bio-inspired architectures demonstrated a rising, bilinear R-curve (Fig. 4d) occurring at approximately halfway through the specimens. We hypothesized that the three toughening mechanisms of crack twisting, crack shielding, and interlocking elicit the rising R-curves of the double-bouligand and bouligand architected concrete.

In the double-bouligand architecture, an initial increase in the R-curve (Fig. 4d) was observed at a specimen depth of approximately 25 mm (Fig. 5a), corresponding to a small load drop ahead of the peak load. This initial toughening was attributed to the bilayer of filaments ahead of the crack tip. Specifically, the filaments in this layer were oriented parallel to the X−Z plane (Supplementary Fig. 7), thereby acting to toughen the material ahead of the crack tip. The second rise in the double-bouligand R-curve occurred at a specimen depth of

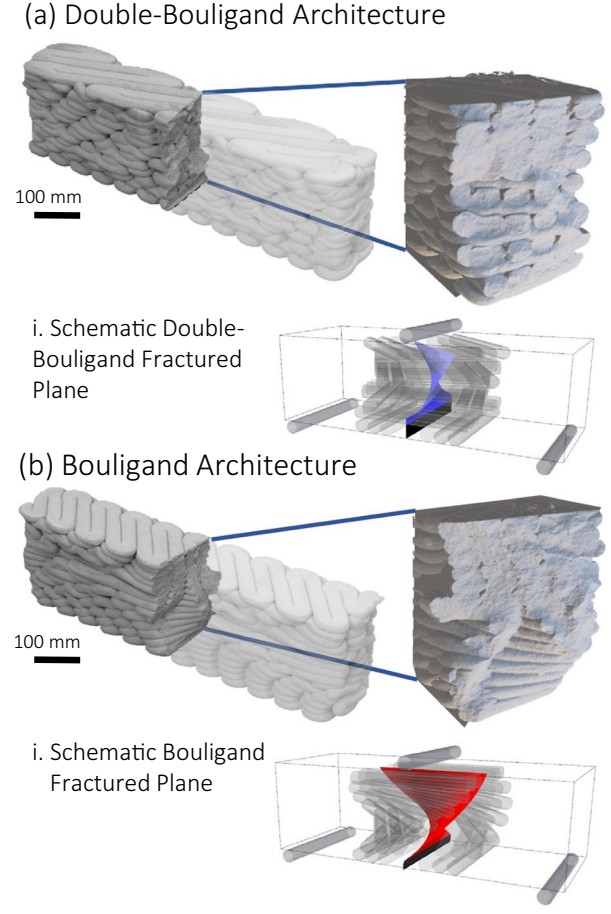

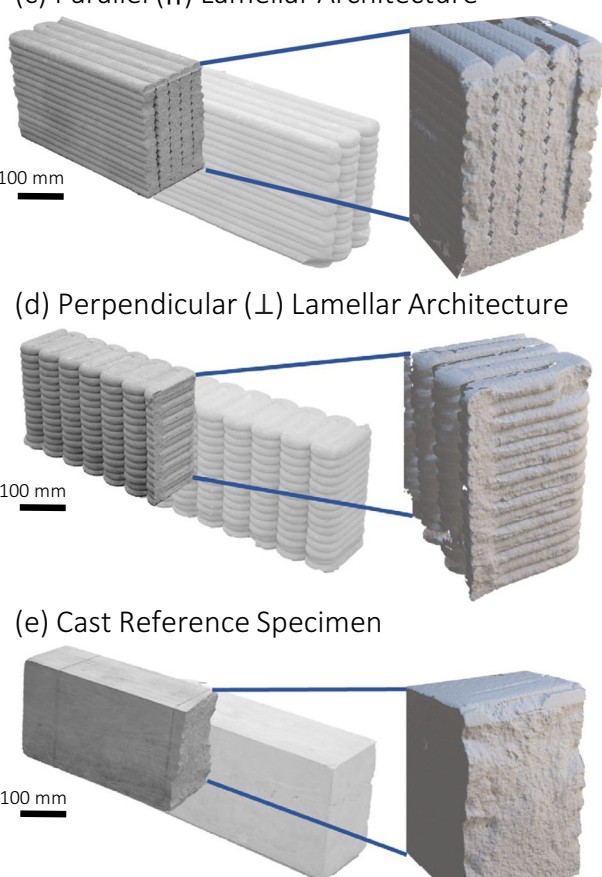

**Fig. 5 | Fractured specimens and fractured planes of architected vs. cast concrete in single-edge notch bend (SENB) test. a** Double-bouligand architecture, and (i) idealized fractured plane, **b** Bouligand architecture, (ii) Idealized fractured plane **c** Parallel lamellar architecture, **d** Perpendicular lamellar architecture, and **e** Monolithic reference cast specimens.

approximately 120 mm. In both the bouligand and double-bouligand specimens, the prominent increase in slope of the respective R-curves was attributed to the mechanical interlocking of the fracture surfaces behind the crack tip.

The crack in the bouligand architecture was highly twisted and extended halfway through the specimen, to a depth of approximately 105 mm measured from the bottom (Fig. 5b). This commensurately marks the depth at which the secondary increase in the slope of the R-curve began (Fig. 4d).

The fracture responses of the double-bouligand and bouligand architectures were distinctly different from the cast and parallel lamellar architectures which exhibited no rise in their R-curve (i.e., a baseline crack initiation fracture toughness). This outstanding fracture behavior clearly indicated that the underlying toughening mechanisms, mimicking those found in the coelacanth fish and mantis shrimp, demand a greater amount of energy to propagate a crack. The cast and parallel lamellar architected specimens only exhibited an initial fracture toughness, as shown by the dashed line in Fig. 4d, and consequently, a brittle fracture. Conversely, the rising R-curve of the bio-inspired architectures suggested that approximately half of the bio-inspired architected specimens remained intact at the onset of toughening.

The toughness of the architected specimens taken from the R-curves was plotted for comparison with the cast reference case (Fig. 4e). The critical stress intensity factor, $K_{Ic}$, or fracture toughness at the onset of cracking, was found to be statistically similar for the parallel lamellar, bouligand, and cast reference cases. The $K_{Ic}$ of the double-bouligand case is statistically lower than the parallel case but statistically similar to the bouligand and the cast reference cases. The $K_{Ic}$, however, captured only the initial fracture phenomena prior to crack propagation that follows, whereas the fracture toughness, $K_{Jc}$, calculated using Eq. (7), captured the additional increase in fracture toughness as the crack extended. The double-bouligand and bouligand architecture demonstrated significantly higher fracture toughness than the cast specimen. Moreover, the double-bouligand architecture represented a statistically similar fracture toughness as the bouligand architecture. Thus, the two bio-inspired concrete specimens demonstrated a higher fracture toughness and a stable crack growth as compared to the cast counterpart, owing to the design of the material's architecture.

The post-peak softening in both the double-bouligand and bouligand specimens was calculated using Eq. (10) considering the entire area under the load-displacement curves of the notched specimens (Fig. 4c). Notably, the significantly higher WOF of the double-bouligand and bouligand architectures (Fig. 4f) indicated the greater amount of energy necessary to propagate a single crack through the bio-inspired architected materials as compared to the perpendicular lamellar architecture, by virtue of the crack path deviating from an otherwise straight fracture plane. Note that the notch tip is located in the filaments in the bottommost layer (which is oriented perpendicular to the x-axis as shown in Supplementary Fig. 7b, d) for both the lamellar and the two bio-inspired cases. This increase in WOF is particularly evident by comparing respectively, the tortuous and twisted fracture surfaces of double-bouligand and bouligand specimens (Fig. 5a, b) to the entirely flat fracture surfaces of the lamellar (Fig. 5c, d) and cast specimens (Fig. 5e).

## Analysis of fractured plane

The double-bouligand specimens (Fig. 5a) exhibited visible crack deflection along the alternating interfaces and through the bilayer filaments up to the first five layers, corresponding to a total theoretical fractured plane rotation of 30° (Fig. 5a–i). Moreover, due to the bilayer arrangement along the twisted plane, there was generally greater crack tortuosity in the fractured surface. This tortuosity is commonly observed in materials having frequent transitions between harder and softer phases[56], which in the double-bouligand architecture are the filament and interface, respectively. Though the total rotation of the fractured surface is less prominent than in the bouligand specimen, the increased tortuosity along the fractured surface contributed to an interlocking effect behind the crack tip which prevented a brittle failure. Therefore, we hypothesized that the presence of the bilayer in the double-bouligand architecture both behind and ahead the crack tip contributed to the higher energy required for fracture compared to the cast specimens. To reiterate, the presence of the bilayer ahead of the crack tip in double-bouligand architecture may be hypothesized to have given rise to the initial increase in the R-curve as an intrinsic mechanism (Fig. 4d) generating a crack shielding mechanism. In short, the predominate toughening mechanism of the double-bouligand architecture can be postulated as the crack deflection (along the interface and through the filament), and crack shielding due to the presence of the bilayer, which behind the crack tip can manifest as frictional interlocking due to the tortuosity along the moderately twisted fractured plane.

For the bouligand specimens (Fig. 5b), the crack path also deflected along the interface between filaments for the first seven layers following the first, notched layer. This corresponded to a total theoretical crack twist angle, $f$, of the fractured plane measured from the notch tip by 70° as defined by a pitch angle, γ, of 10° (Fig. 5b–i). The secondary increase in the slope of the bouligand R-curve (Fig. 4d) at a depth of 105 mm suggested that the crack deflection, may have triggered crack shielding behind the crack tip in the form of mechanical interlocking along the twisted, fractured surface. Therefore, it can be postulated that the crack twisting and interlocking mechanisms were associated with the toughening behavior in the load-displacement response of notched bouligand specimens.

The R-curve slope of the bouligand specimen beginning approximately halfway through the sample is higher than the double-bouligand architecture due to a greater twisting of the fractured plane which gives rise to more prominent mechanical interlocking. It is hypothesized that greater crack twisting in the benchmark bouligand architecture increases interlocking and leads to more effective toughening. This interlocking is only achieved in architected specimens and was not observed in cast counterparts. In addition, it is observed that a certain degree of crack twisting was required for either of the two architected materials before any interlocking mechanism could engage. This is deduced from the initially rapid crack extension, illustrated by a nearly flat R-curve slope in bouligand architecture and the flat middle region of the R-curve slope in the double-bouligand architecture.

## Theoretical analysis of critical energy release rate

Deflecting a crack away from the direction of applied load in Mode-I is a challenging task in brittle and quasi-brittle cementitious materials[57]. Crack twisting, also considered as the summation of several incremental crack deflections, requires the crack tip to undergo a localized in-plane shear (Mode-II)[58]. It is well-understood that fracture under shear (Mode-II) is a more energy intensive than the initial fracture under tension (Mode-I) as commonly reported in cementitious materials[57]. Presence of shear failure thus increases the critical energy release rate, $G_c$, that defines the fracture toughness of the specimen.

In this work, a twisted fracture path was observed in the bouligand architecture (Supplementary Note 3 and Supplementary Fig. 3a). The increase in the critical energy release rate, $G_c$, due to crack twisting can be analytically illustrated for a fracture path following along a defined pitch angle, γ. The analytical solution was based on linear elastic fracture mechanics (LEFM) and hence is limited in capturing the toughening mechanisms that occur ahead of the crack tip in the quasi-brittle concrete. Nonetheless, LEFM provides insight into the toughening mechanisms of helically designed concrete.

In the bouligand architected concrete studied here, the twist angle, $\phi$, can be computed as the summation of pitch angles, $\gamma$, at each 3D-printed layer. The kink angle, $\alpha$, can be computed as the angle between the z-axis and the projection of the z-axis onto the original Z-X plane (Supplementary Fig. 3b)[59]. In this manner, the twist angle, $f$ reaches 90° at a depth of 117 mm though the bouligand specimen (with fifteen layers and $\gamma = 10°$), which is the maximum possible rotation of the fractured plane (Supplementary Fig. 3c). At this given maximum rotation of 90° (Supplementary Fig. 3d), the resulting critical energy release rate along the depth of the specimen, $G_c$, in comparison to the critical energy release rate of the material itself, $G_c^m$, if realized can increase by more than a factor of 12[59]. This total ideal rotation, however, is not entirely realized in the bouligand concrete specimen. Instead, the partial twisting that occurs in the first eight layers would increase the critical energy release rate of the fracture, thereby increasing the reported fracture toughness by 63% and 88% in the double-bouligand and bouligand specimens, respectively, compared to the cast counterpart.

In the case of weaker interfaces, $G_c/G_c^m$ would be lower (Supplementary Fig. 3d) for a given pitch angle, indicating reduced crack twisting and Mode-II contribution. This lower ratio can then be compensated for by increasing the pitch angle in the design to maintain the interaction of a pre-existing crack with the weaker interface.

## Role of the weak interface and toughening mechanisms

Using micro-CT, we hypothesize the macroscopic pores (Supplementary Note 4, Supplementary Fig. 4) in the horizontal interfaces between the filaments in the additively manufactured architected specimens promoted the predominant toughening mechanisms, specifically crack deflection in double-helical concrete and crack twisting in helical concrete, by providing preferential crack pathways. In addition, a higher degree of porosity observed in the vertical interface between printed filaments, shown in Supplementary Fig. 4c, f, may further promote crack propagation in double-twisted and twisted arrangements along the filament interfaces.

Concrete, as with most cementitious composites, is known to have low tensile strength and fracture toughness (Fig. 6)[13]. However, engineering toughening mechanisms into concrete, by using the double-bouligand and bouligand architectures, has led to

improvements in the fracture toughness as compared to cast counterparts. This enhancement is benchmarked using an Ashby Plot against reference cementitious mortars as well as other classes of materials including non-technical ceramics, polymers, and glasses (Fig. 6).

In contrast to engineered composites, nature has developed a plethora of tough and strong biological composites through the architected arrangement of modest materials. The architected themes in biological materials rely on the presence of weak interfaces to incite the underlying mechanisms that give rise to enhanced toughness[4]. However, engineering the toughening mechanisms into construction materials such as concrete requires an understanding of both fracture and advanced manufacturing processes. The weak interfaces and internal heterogeneities in additively manufactured concrete are widely studied[16,60–62], such as those discussed in Supplementary Fig. 4, but not commonly exploited. This presents a perfect opportunity to harness the processing-induced flaws in favor of material with crack-resistant characteristics.

The tensile strength of concrete poses a significant limitation in the design of concrete structures. This holds true even for structures in which the principal stresses are primarily compressive, as localized tensile stresses can result in cracking, which leads to spalling and deterioration[13], and in extreme cases, progressive collapse[63]. These mechanisms of damage are exacerbated in additively manufactured concrete, due to the presence of weak interfaces between filaments. Improving concrete's susceptibility to cracking can be addressed by increasing its resistance to fracture, first and foremost in Mode-I[64]. The purposeful design of concrete via large-scale additive manufacturing technology can provide a pathway to address the poor fracture toughness under tension (Mode-I) in conventional concrete. This approach may allow design strategies for unreinforced concrete structures that could rely on fracture resistance under tension. By enhancing the tolerance to cracking, adopting purposeful architectures can help increase the service life of concrete and other quasi-brittle infrastructure materials.

In short, the non-brittle performance of the double-bouligand and bouligand architected materials are engineered in this work. Crack deflection, shielding, and twisting enabled by purposeful designs and robotic additive manufacturing, provide an alternative to conventionally cast counterparts as well as rectilinear additively manufactured concrete in which the improved fracture properties cannot be achieved. This fosters an approach towards the design of non-brittle construction material without necessarily requiring the addition of fibers or reinforcement.

In conventional concrete design, cracked concrete sections are not taken into account for reinforced concrete structures, owing to their inherent lack of fracture resistance. In this work, we address the long-standing limitation of concrete's tensile properties, which govern the design of concrete structures. The increase in fracture resistance in Mode-I translates to a potential for relying on concrete's tensile capacity, which goes beyond our conventional understanding of how concrete is typically designed and constructed solely in compression. By developing design schemes that inherently improve fracture toughness, we aimed to enhance concrete's resistance to cracking, which can also lead to improved long-term performance where cracks can be initiated due to physical or chemical degradation. This study highlights the difference between conventional brittle fracture in cast concrete and the progressive resistance to cracking in the proposed architected materials. Given the growing global interest in digital fabrication and advanced manufacturing technologies at a large scale, the development of opportunistic approaches to the design of materials and structures can help overcome the limitations of the mechanical properties of concrete. This study investigates the fracture toughness of architected materials at a component level. The investigation of fracture toughness and the role of internal flaws at larger

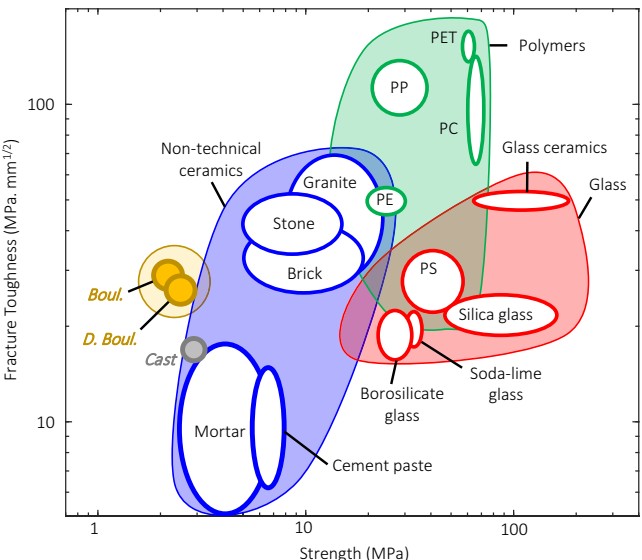

**Fig. 6** | Ashby Plot demonstrating the fracture toughness vs. strength of polymers[66,67], glass,[66,67] non-technical ceramics[66,68], cement paste[69,70], and cementitious mortar[55,71–73] benchmarked against the cast materials and the double-bouligand and bouligand architected counterparts studied in this work.

scales are equally important and should not be discounted in the ongoing effort to build larger additively manufactured structures. Future studies can focus on engineering the interfacial properties using two-component extrusion through specific interfacial chemistry. This approach can also allow for the experimental engineering of interfacial properties relative to filament properties, in a gradient fashion, enabling the tailoring of local interfacial characteristics to enhance global properties. Recently developed frameworks have shown great utility in examining broader material, interfacial, and geometric domains, such as coupled phase-field and cohesive zone modeling frameworks.[65] A numerical approach to these investigations can enhance the experimental method to advance our understanding of engineering the fracture response in brittle and quasi-brittle layered or architected materials.

## Methods

### Algorithms for architecture-specific toolpath generation

A conventional toolpath algorithm slices a user-defined geometry at uniform increments to form elevation contours that are then automatically filled with a generic infill pattern. Architecture-specific toolpaths work in reverse, by beginning with a purposeful arrangement of extruded filaments to accurately emulate a desired architecture. The desired arrangement forms a base architected unit (e.g., a line, polyline, curve, or series of points), which along with the user-defined input geometry, the filament dimensions, and the printing speed, are necessary input parameters for the Grasshopper toolpath algorithm developed here (Supplementary Fig. 5a). The base architected unit (Supplementary Fig. 5b) is defined here as a continuous polyline. It is then arrayed in the desired fashion (e.g., helically) in all cartesian directions to generate an infill volume (Supplementary Fig. 5c). The user-defined input geometry, represented as a surface, mesh, or boundary representation (BRep) is then used to split and cull the extraneous arrayed volume (Supplementary Fig. 5d). Endpoints from the remaining polylines are extracted and sorted in series. As with the conventional slicing algorithm, the start and end of the polyline are fitted with an entry and exit point from which the end-effector will begin and end the toolpath (Supplementary Fig. 5e). The final polyline is again discretized into points of which the cartesian coordinates (Supplementary Fig. 5f–i) are used to define the Targets of the Move instructions in RAPID code. Additional detail about architecture-specific approach to toolpath generation can be found in Supplementary Note 5 and Supplementary Fig. 5 in contrast to conventional toolpath generation (Supplementary Note 6 and Supplementary Fig. 6).

### Fabrication of architected concrete

The architecture-specific toolpath algorithm was used to design concrete specimens with both bouligand and double-bouligand architectures (Supplementary Fig. 7a, Supplementary Videos 1–4). The specimens had dimensions of 130 × 200 × 700 mm and a pitch angle of $\gamma = 10°$ for both bouligand and double-bouligand architectures (Supplementary Fig. 7b). The pitch angle was chosen based on the preliminary results on optimal angles found for similar tests on smaller 240 × 60 × 60 mm concrete specimens using a smaller robot arm. The specimens were fabricated with the 2-K process (Supplementary Fig. 7c). The fabricated samples (Supplementary Fig. 7d) were notched for the single-edge notch bend test and otherwise remained unnotched for the three-point bend test. For both bio-inspired cases, the filaments in the bottommost layer were oriented perpendicular to the x-axis (Supplementary Fig. 7b, d). Two additional lamellar architectures, denoted as Parallel (‖) and Perpendicular (⊥) relative to the specimen length along the x-axis, were fabricated to characterize the fracture toughness of the filament and interface, respectively. The extruded concrete filaments were approximately 27 mm wide and 13 mm tall. Conventionally cast specimens of equal dimensions were

fabricated to serve as a reference case. The beams were then covered with plastic and maintained an average relative humidity of 81 ± 5% for 7 days of curing until tested.

### Material composition and processing parameters

The concrete mixture was comprised of 869.9 kg/m³ of fine aggregates, 588.8 kg/m³ of CEM I 52.5 R cement, 231.5 kg/m³ of water, and 11.5 kg/m³ of BASF MasterRoc SA 167 alkali free set accelerator. The concrete was extruded at a rate of 2.0 L/min and dosed with accelerant at a rate of 0.012 L/min. Thus, a water-to-cement ratio of 0.39, a fine aggregate-to-cement ratio of 1.47, and an accelerant dosage of 0.02% per mass of cement were used. The fine aggregate particle size distribution followed the upper bound of ASTM C33 limits with a maximum particle size of 4 mm and the smallest particle size of 0.2 mm. The nozzle speed was maintained at 60 mm/s ± 5 mm/s during the 2-K fabrication process. The concrete pressure inside the nozzle mixing chamber ranged from 50 ± 10 kPa and the temperature exiting the nozzle ranged from 35 °C ± 1 °C. These parameters were held constant during the fabrication of each specimen to maintain material consistency in each and across fabrication sessions. Each session comprising the fabrication of 8 architected elements took approximately 90 minutes, leading to a production rate of roughly 0.012 m³/hour (Supplementary Note 2).

### Characterization of mechanical and fracture properties and statistical analysis

Load-displacement plots of unnotched and notched specimens using a mechanical testing unit are demonstrated in Fig. 3a, c, respectively. The instantaneous load and displacement were measured at a frequency of 10 Hz. Various mechanical properties were calculated based on the measured load and crosshead displacement, including the modulus of rupture (MOR), fracture toughness $K_{Jc}$, and the work-of-fracture (WOF), as illustrated in Fig. 3b, e, f, respectively. The notch tip in the notched specimen was positioned in the filaments in the bottommost layer (which is oriented perpendicular to the x-axis as shown in Supplementary Fig. 7b, d) for both the lamellar and the two bio-inspired cases (Supplementary Note 7). For all measurements of mechanical properties, the plotted error bars represent the standard deviation in the specimen data. The statistical significance of the sample was calculated based on two-tailed t-tests, by first testing the statistical similarity of variance using an f-test and then testing the statistical significance between the sample means. A significance level of 0.05 for the computed p-values was used to determine the significant differences in the results. Six samples for double-bouligand, six samples of bouligand, four samples of cast, and two samples of perpendicular and parallel cases were tested for the calculation of fracture toughness (R-curve, WOF, $K_{Jc}$). Four samples for double-bouligand, four samples of bouligand, four samples of cast, and two samples of perpendicular and parallel cases were tested for the calculation of MOR. The calculation of the modulus of rupture, fracture toughness, and work-of-fracture are further discussed in Supplementary Note 8.

### Characterization of material's fresh and hardened properties

All early-age and mature properties are summarized in Supplementary Note 9. Early-age properties of the concrete mixture upon extrusion are reported in Supplementary Fig. 8. Isothermal calorimetry was used to characterize hydration and the use of accelerant. Cone penetration tests were conducted to evaluate rheological property (yield stress). Fresh density immediately after extrusion was also measured. Mature properties of the same concrete mixture (i.e., cast in molds using the same extrusion process) were also reported in Supplementary Fig. 8 including 7-day compressive strength, 7-day Young's Modulus (tested at a rate of 0.6 mm/min following ASTM C39 standards), and hardened density. Additional details on each experiment can be found in Supplementary Note 9.

## Data availability

Source data are provided with this paper. Relevant raw data is also provided with this paper. The data generated in this study has been deposited in Data Commons (https://doi.org/10.34770/7d7m-db31) including raw and processed data.

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

## Acknowledgements

The authors would like to thank the Civil, Mechanical, Manufacturing Innovation (CMMI) Division of the National Science Foundation (NSF) for the support of the robotic manufacturing aspects of this research through the Advanced Manufacturing Program (No. 2217985) and the partial support of the bio-inspired aspects through Engineering for Civil Infrastructure Program (No. 2129566, Collaborative). The authors would like to thank Dr. Aimane Najmeddine, Mahsa Rabiei, Dana Daneshvar, and Dr. Lara Tomholt for participating in the robotic experiments in the revision or the collective discussion of the findings.

## Author contributions

H.S. and R.M. identified and conceptualized the idea, formalized the research questions, R.M. and H.S. designed the experiments and led the robotic additive manufacturing experiments, A.P. designed the robotic manufacturing toolpath in Grasshopper, S.G. led the extrusion, sensing, monitoring, and tuning processing parameters during robotic manufacturing, and conducted the image and data analysis, A.P., S.G., H.S., and R.M. participated in the fabrication and mechanics experiments, A.P., S.G., and R.M. prepared the initial manuscript, S.G. and R.M. carried out the revisions and additional mechanics experiments and edited the manuscript. All authors contributed to the discussions.

## Competing interests

The authors declare that they have no known competing financial interests or personal relationships that could have appeared to influence the work reported in this paper.
