## [Peer Review File · Nature Communications]

Tough double-bouligand architected concrete enabled by robotic additive manufacturingREVIEWER COMMENTS

Reviewer #1 (Remarks to the Author):

The paper presents a pioneering architected and additively manufactured cementitious material with great potential in future explorations of this technology since it is inspired by the very effective double-bouligand architecture. The holistic approach, the educative coverage of the state-of-the-art and the impressive illustrations demonstrate the breakthrough research outcome.

The research conceptualization is unique, however directly linked to relevant research on other materials, such as of CFRP. Can you highlight the additional challenges induced by the cementitious matrix nature? Based on supplementary materials, it is evident that the layers' thickness is very large compared to the sample dimensions, therefore a scaling issue arises. This is not the case for CFRP and other additively manufactured materials, eg. ceramics. Can you comment on the essential differences among materials?

The toughening onset, as defined in line 159, is not clarified. What is considered as softening starting point? How can you pinpoint that this phenomenon starts at 13% of peak load, since no indication of slope variation is evident. Can you please explain further this fundamental measured variable?

The architected material is characterized as 'concrete' throughout the text, however, only cement paste is printed eventually. Please, revise this misunderstanding and revise accordingly the used terminology.

The compressive strength is not reported, however this information remains the main criterion for materials selection and design in a more conservative approach.

It will be interesting to induce artificially debonded layers among stacks, this way trigger loss of bond between stacks of the same unit and stacks placed in a different angle. This way, the effect of porous interfaces on the energy dissipation could have been tracked.

Why the pitch angle remains constant at different locations among the height of the beam? For instance, why not considering different angles at the tensile and compressive zones of the sample?

Grammar and syntax errors are limited, however a thorough revision is recommended (e.g. with advanced with...line 118).

Reviewer #2 (Remarks to the Author):

Structures are elementary foundations of life and the way we build is part of our social developments. In pre-industrial times, only natural materials were used for building, essentially natural stone, wood and clay. Knowledge of the natural properties of materials was crucial in shaping materials and assembling them into structures like the Gothic cathedrals or cities made of clay, like Shibam in Yemen. With industrialization, it became possible to create new materials artificially, and in construction, steel and concrete became the main building materials. However, the way of forming and joining with these artificial materials is still craft-based and often does not take advantage of the materials' capabilities.

This manuscript is based on the innovative technology of additive manufacturing to shape cementitious materials. Additive Manufacturing technologies in construction (AMC) have generated tremendous global research interest in recent years, and are particularly ideal for concrete construction to remove the limited shaping provided by formwork. The construction industry has also recognized the potential of so-called 3D concrete printing (3DCP), and the first proto-type constructions have already been realized in various countries. Here, the technology of concrete extrusion is mainly used in a 1-component system based on 3-axis gantry systems, where the cementitious material is often applied layer by layer. In summary, in contemporary 3DCP three-

dimensional objects are printed, but the design is still based on the routines of two-dimensional traditional construction processes, such as laying bricks in layers.

To take advantage of the form freedoms of AMC, it is essential to understand material, manufacturing process, and design as a unit. The authors bring a compelling innovative new approach of unifying material, process and form: Using a 2-component robotic additive manufacturing process, cementitious material is deposited and structurally arranged in a novel three-dimensional pattern. The design of the arrangement is inspired by nature, the double-bouligand architecture found in the coelacanth fish. In contrast to the efforts made in extrusion at the micro level of the material to achieve a bond that is as homogeneous as possible across the layers, the authors turn the supposed problem into an advantage and create the bond at the meso-scale via a three-dimensional, geometrically complex interlocking of the layers. The authors hypothesize that in nature, architectural material design is crucial to toughness and this can be achieved through the complex arrangement of layers in 3DCP. The novelty of the present approach is not to adopt natural construction as a formal design principle, but to capture natural design as a construction principle with its entire complexity and to map it as far as possible in the entire process chain of material technology, printing process and structural design.

The particular process components for controlling this complex process are:

1. A two-component AM process, allowing real-time tuning of the material's rheological properties through the acceleration or deceleration of hydration at the nozzle.
2. Sensor-controlled mixing, pumping and printing through measuring the pressure and temperature of the wet matrix in the different process steps to evaluate the material quality and operational performance.
3. A Grasshopper-based architected toolpath generation algorithm.

The authors impressively demonstrate that this novel approach at the meso level, on the one hand, significantly expands the possibilities of additive manufacturing technology in construction and, on the other hand, enhances component behavior. The double-bouligand designs are benchmarked against bouligand, conventional rectilinear counterparts and monolithic cast. The hypothesis that the spatial arrangement of the proposed complicated double-Bouligand architecture of concrete materials can enable favorable mechanical properties is impressively confirmed by the tests performed and the methods used.

The work is of extraordinary relevance to structural design and architecture. It is an independent and highly innovative contribution to advance the adoption of AMC and to inspire engineers and architects for designs with a novel "logic of form" based on material and resource efficiency. Data analysis, results and conclusions are valid. The chosen methods are convincing and described in detail, the reproducibility is given. The supplementary information is very good and enriches the main work.

I recommend the acceptance of the paper.

23/07/09

Harald Kloft

Reviewer #3 (Remarks to the Author):

The paper describes a study to determine the effect of an architected material composition achieved by 3d printing in bouligand and double bouligand architectures on tensile strength and fracture toughness of cementitious composites, with the aim of increasing particularly the latter and thereby overcoming a significant drawback of the use of these materials in building structures. The approach is inspired by architected materials found in nature.

Although this concept is innovative in its consideration of material properties and use of 3d printing methods to achieve them, I have to advise against publication of the manuscript in its current form, for several reasons.

First of all, the paper is based on very little data. Unfortunately, it is not explicitly stated how many specimens were tested. The supplementary data just mention 'minimum of 2 per specimen type'. In my opinion that is too little to warrant any solid conclusion or perform any statistical analysis, it can merely point into a direction. Considering the procedure is not overly complicated, a

considerably higher number of experimental results would be expected for a publication in this journal.

Furthermore, besides the destructive testing results, there is very little reporting on supplementary measurements, properties, etc. There is no data on the fresh or hardened state properties of the printing mixture, no data on interface or bulk material strength, material density or object density (only some very limited data in the supplementary notes file), etc. Some hardened state properties used in analyses are apparently based on 28-day testing, while the specimens themselves were tested at 7 days. Considering the use of accelerators, it is not evident they can be accurately calculated from each other. The effect of printing vs casting for these properties is not discussed. And the actual numbers on these properties are not given.

The printing quality of the specimens would generally be considered sub-standard due to the large air voids (visible in the pictures, as well as in the supplementary data). This raises the question how applicable the results would be in general, as most printing operators would not print this way for a variety of reasons.

The printing method also results in specimens of different geometries. Although all roughly a solid beam, the irregular surfaces may influence the effective sizes used in calculations as well as the local stress distributions. It is also not clearly indicated where the notches are applied respective to the interfaces, and why.

The experimental procedure can also be questioned. A 3-point bending set-up is not suitable to compare and analyse post-fracture behaviour if the fracture does not remain in the area under the loading point – which it does not for the (double-)bouligand architectures – because the stress levels reduce fairly quickly when moving away from the loading point. A set-up featuring larger areas of constant tensile stress would have been more suitable (4p bending or ideally uni-axial tension). The applied displacement-controlled loading rate is very low, favouring crack path diversions. However, at higher (more common) rates the observed differences may disappear. At the very least, this should be discussed, and preferably also studied experimentally.

The relevance of increasing the fracture toughness of (printed) concrete this way is questionable. It seems to come at the cost of tensile strength and stiffness. Particularly the latter is a very important parameter for structural concrete engineering. The tensile is generally considered less important as we cannot rely on the tensile strength of concrete anyway. In practice we should always consider that the concrete may be fractured due to other causes such as shrinkage, creep, and (related) forced deformations. But also the relevance of fracture toughness should for structural concrete should not be overestimated. In structural engineering we typically consider loads to be (quasi) static, thus the energy dissipation in fracture becomes less important: something either breaks or not. Fracture toughness would only be relevant in dynamic or (short duration) temporary loadings such as explosions or earthquakes. But even then, it can be debated (and should be analysed in such a paper) whether the obtained increase in fracture toughness is relevant for such cases. Because although the authors claim the developed material design methodology can obviate the need for fibers or reinforcement (bars), I expect the energy absorption capacity of such additions still to be many times higher than the fracture toughness found in these architected materials.

Owing to the small number of tested samples and tests performed, not much can really be concluded from the work. This is perhaps the reason why several textual formulations appear repeatedly (basically the conclusion that a (double-)bouligand architected material structure improves the fracture toughness).

The paper, finally, would need some restructuring, as the method comes at the end (should be before results), the method is described much too generally, and there are no conclusions (actually the discussion seems to hold the conclusions, but this leaves for a very brief discussion that could have gone more in depth).

Please refer to the uploaded reviewer files for further detailed comments.

REVIEWER COMMENTS

Response. We appreciate the significant review and major suggestions that have helped us improve this manuscript. After conducting additional experiments and statistical analysis and updating the results, the findings with regard to improved fracture toughness from the proposed double-bouligand architecture have remained unchanged while any other changes have been reported and manuscript have been revised accordingly. A point-by-point response has been provided and in most cases, major corresponding changes in the manuscript and supplementary information.

Reviewer #1 (Remarks to the Author):

The paper presents a pioneering architected and additively manufactured cementitious material with great potential in future explorations of this technology since it is inspired by the very effective double-bouligand architecture. The holistic approach, the educative coverage of the state-of-the-art and the impressive illustrations demonstrate the breakthrough research outcome.

Response. We appreciate such an understanding of the paper at the research and state-of-the-art level.

The research conceptualization is unique, however directly linked to relevant research on other materials, such as of CFRP. Can you highlight the additional challenges induced by the cementitious matrix nature? Based on supplementary materials, it is evident that the layers' thickness is very large compared to the sample dimensions, therefore a scaling issue arises. This is not the case for CFRP and other additively manufactured materials, e.g. ceramics. Can you comment on the essential differences among materials?

Response. Thank you for the opportunity to clarify. We have revised the manuscript and discussed the additional challenges of layer-wise manufacturing with concrete compared to carbon-fiber-reinforced plastic (CFRP) material, including scaling effects in **Supplementary Information Lines 150 - 155**. The essential differences between architected concrete vs. other systems (CFRP, ceramics) are as follows. We have revised the manuscript accordingly in **Supplementary Information Lines 150 - 159**:

- **Challenges in cementitious materials:** In extrusion-based additive manufacturing of hydraulic suspensions (e.g., cementitious), additional challenges include the presence of large early-age deformation (partly due to the large size of the printed filaments)^[1] and reliance on hydration at early stages^[2] for the hardening process. Numerous studies have been conducted to engineer the rheological and hydration characteristics of additively manufactured cementitious materials to address these two challenges.^[3,4,5,6]
- **Scaling differences:** As for the scaling and relative size of the filament vs. the structure, one of the challenges in scaling up the AM technology is the large sizes needed in civil

¹ Moini, R., *et al.*, 2022. *Cement and Concrete Composites*, 131, p.104538.

² Moini, R., *et al.*, 2021. *Cement and Concrete Research*, 147, p.106493.

³ Douba, A., *et al.*, 2020. *Digital Concrete 2020 2* (pp. 32-41). Springer International Publishing.

⁴ Douba, A., *et al.*, 2022. *Cement and Concrete Composites*, 125, p.104301.

⁵ Das, A., *et al.*, 2022. *Cement and Concrete Research*, 162, p.107004.

⁶ Marchon, D., *et al.*, 2018. *Cement and Concrete Research*, 112, pp.96-110.

infrastructure and the balance between the large size of the structures and the production/fabrication rate. To address these challenges, large nozzle sizes have been widely developed in recent years.^[7] Previous works have discussed the effect of the size of the structure on early-age deformation and yield stresses required for a given production rate.^[8] In addition, the effect of nozzle size on production rates has also been more carefully studied in recent years, indicating the smaller the nozzle size, the longer the production time. Two-component extrusion systems have been developed to enhance the production rate, especially in the vertical direction, through acceleration of the hydration of cementitious material at the nozzle tip. The contribution of such acceleration to overall production rates has been discussed in recent studies from a processing perspective. The findings have highlighted that the use of a two-component system greatly facilitates printing at scale even within 24 hours benchmark.^[7] In addition, from a purely mechanics perspective, size effects are in play as the structures scale up, which are influenced by the interplay between the structure size and filament (or interface flaw) size. Those size effects would also be in play in CFRP if those structures were to be scaled up. Additional discussion is also provided on the size in response to Reviewer 3.

- In short, in terms of the ability to scale up, it is beneficial to the production rate to work with larger filament sizes (27 mm) and a 2K system. Here, the 2K system, was mainly used to achieve high-fidelity geometric accuracies for achieving refined architected materials, although some examples of high vertical production rates were given in Fig. 3. This is also advantageous from mechanical properties and design of mesoscale architecture perspective, to have control over the feature sizes (solids or pores) within a reasonable fabrication time (report from paper our production or flow rate). As such, in contrast to CFRP additive manufacturing, here one of the key differences is the sizes of the final components in the civil infrastructure components. A short discussion is also highlighted in the Supplementary Information Line 160 - 166.

The toughening onset, as defined in line 159, is not clarified. What is considered as softening starting point? How can you pinpoint that this phenomenon starts at 13% of peak load, since no indication of slope variation is evident. Can you please explain further this fundamental measured variable?

Response. We are not defining softening as the onset of toughening or change of slope in load-displacement. The fundamental toughening mechanisms are discussed in terms of crack twisting and interlocking. In Fig. 4c, we are merely reporting the presence of a softening at the tail of the load-displacement plots in the double-bouligand and bouligand cases, which is absent in the cast. The post-peak ‘softening point’ is *the beginning of the gradual decrease in the load after the peak*. Please see Main Line 159.

The architected material is characterized as ‘concrete’ throughout the text, however, only cement paste is printed eventually. Please, revise this misunderstanding and revise accordingly the used terminology.

Response. We did not print with cement paste. We have used cementitious mortar as elaborated in the section “Material Composition” in the Main manuscript (Line 395 – 401) with a maximum aggregate size of 4 mm.

⁷ Wangler, T., et al., 2022. *Cement and Concrete Research*, 155, p.106782.

⁸ Roussel, N., 2018. *Cement and Concrete Research*, 112, pp.76-85.

The compressive strength is not reported, however, this information remains the main criterion for materials selection and design in a more conservative approach.

Response. We have conducted new experiments and have reported the 7-day compressive strength results for the material extruded using the same process. This is added in the revised Supplementary Information Fig. 8 and Line 392 - 395. This may facilitate the commonplace conservative approach, as indicated by the reviewer, to develop new types of design of architected concrete.

It will be interesting to induce artificially debonded layers among stacks, this way trigger loss of bond between stacks of the same unit and stacks placed in a different angle. This way, the effect of porous interfaces on the energy dissipation could have been tracked.

Response. The focus of the current manuscript was to investigate the role of the currently 'weak' interface when coupled with double-bouligand architecture on the induced interesting toughening mechanisms and potential improvements in resistance to cracking. Nevertheless the proposed notion is a great idea. Indeed, we are exploring this in our next study to artificially weaken or strengthen the interfacial properties using the two-component extrusion through specific types of interfacial chemistry and functionally grade (increase or decrease) the interfacial properties relative to the filament properties. These types of "chemically-weakened interfaces" using a secondary interlayer material in a fresh state are the subject of an entirely new study^[9] using two-component extrusion and may be beyond the capacity of this paper. These types of interfacial properties can be approached experimentally in a gradient fashion to engineer the local interfacial properties and tailor the global properties and can be hypothesized to enhance interfacial energy dissipation and overall fracture toughness.

We are also numerically investigating this question to investigate the role of interfacial energy dissipation on the overall fracture response of brittle layered 'hard-hard' materials in a study that is currently under review.^[10] In this entirely new numerical framework recently developed in our lab, phase-field (to capture fracture in the layered bulk) was coupled with a cohesive zone model (to capture fracture in the interface). The framework was validated against linear elastic fracture mechanics theory for a crack impinging on an interface in bilayer hard-hard materials for crack penetration and crack deflection conditions (dominated by the ratio of fracture toughness of interface relative to fracture toughness of the bulk). The framework can be extended for various boundary conditions and mesoscale multi-layered 3D-printed and architected materials, although it requires further investigation to extend it from brittle to quasi-brittle systems in our next study. We can examine the effect of 'weakening' the interface numerically in bilayers but this may take a significant space and effort beyond the patience of this study. Most certainly, as suggested by the reviewer, new directions in tuning interfacial properties will be pursued.

We have further discussed these studies to contextualize the future steps in the current manuscript in Main Line 354 – 362.

From a mechanics perspective as elucidated via Supp. Fig. 3, we can at least expect that intentionally weaker interfaces would lead to a lower G_c/G_c^m (Supplementary Fig. 3d) for a given same pitch angle, indicating a lower crack twisting and mode II contribution. Such a lower ratio

⁹ National Science Fund (NSF), 2026, Award No. 2217985.
https://www.nsf.gov/awardsearch/showAward?AWD_ID=2217985&HistoricalAwards=false

¹⁰ Najmeddine, A. et al., Available at SSRN: <https://ssrn.com/abstract=4783033>

can be balanced by a higher pitch angle to maintain the interaction of a pre-existing crack with the weak interface. These phenomena are further contextualized in the Main Line 287 – 290.

Why the pitch angle remains constant at different locations among the height of the beam? For instance, why not considering different angles at the tensile and compressive zones of the sample?

Response. This is also a great idea to use the gradient of pitch angles. Given the current study aimed to understand the presence of a double-helical pitch angle, we mainly focused on comparison against helical, as well as layered, and cast cases, in this study. In other words, we designed the experiment to investigate the interaction of a pre-existing crack with the given design of weak interfaces. Past studies have shown the effect of bouligand design on weak interfaces under compression.^[11] However, here, the SENB tests have been used to probe Mode-I (opening or tension) fracture in which the crack tip remains under tension. Nevertheless, we have also theoretically contextualized the presence of shear (Mode-II) and how, in the engineering of the pitch angle, the mixed mode could be considered (as discussed through the ratio G_c/G_c^m in Fig. S3). We have revised the conclusion to briefly discuss some of these future directions. Please see the Main Line 356 – 362.

Grammar and syntax errors are limited, however, a thorough revision is recommended (e.g. with advanced with...line 118).

Response. Thank you for this suggestions. We have revised accordingly.

Reviewer #2 (Remarks to the Author):

Structures are elementary foundations of life and the way we build is part of our social developments. In pre-industrial times, only natural materials were used for building, essentially natural stone, wood and clay. Knowledge of the natural properties of materials was crucial in shaping materials and assembling them into structures like the Gothic cathedrals or cities made of clay, like Shibam in Yemen. With industrialization, it became possible to create new materials artificially, and in construction, steel and concrete became the main building materials. However, the way of forming and joining with these artificial materials is still craft-based and often does not take advantage of the materials' capabilities.

This manuscript is based on the innovative technology of additive manufacturing to shape cementitious materials. Additive Manufacturing technologies in construction (AMC) have generated tremendous global research interest in recent years, and are particularly ideal for concrete construction to remove the limited shaping provided by formwork. The construction industry has also recognized the potential of so-called 3D concrete printing (3DCP), and the first proto-type constructions have already been realized in various countries. Here, the technology of concrete extrusion is mainly used in a 1-component system based on 3-axis gantry systems, where the cementitious material is often applied layer by layer. In summary, in contemporary 3DCP three-dimensional objects are printed, but the design is still based on the routines of two-dimensional traditional construction processes, such as laying bricks in layers.

To take advantage of the form freedoms of AMC, it is essential to understand material, manufacturing process, and design as a unit. The authors bring a compelling innovative new approach of unifying material, process and form: Using a 2-component robotic additive

¹¹ M. Moini, Purdue University, 2020.

manufacturing process, cementitious material is deposited and structurally arranged in a novel three-dimensional pattern. The design of the arrangement is inspired by nature, the double-bouligand architecture found in the coelacanth fish. In contrast to the efforts made in extrusion at the micro level of the material to achieve a bond that is as homogeneous as possible across the layers, the authors turn the supposed problem into an advantage and create the bond at the meso-scale via a three-dimensional, geometrically complex interlocking of the layers. The authors hypothesize that in nature, architectural material design is crucial to toughness and this can be achieved through the complex arrangement of layers in 3DCP. The novelty of the present approach is not to adopt natural construction as a formal design principle, but to capture natural design as a construction principle with its entire complexity and to map it as far as possible in the entire process chain of material technology, printing process and structural design.

The particular process components for controlling this complex process are:

1. A two-component AM process, allowing real-time tuning of the material's rheological properties through the acceleration or deceleration of hydration at the nozzle.
2. Sensor-controlled mixing, pumping and printing through measuring the pressure and temperature of the wet matrix in the different process steps to evaluate the material quality and operational performance.
3. A Grasshopper-based architected toolpath generation algorithm.

The authors impressively demonstrate that this novel approach at the meso level, on the one hand, significantly expands the possibilities of additive manufacturing technology in construction and, on the other hand, enhances component behavior. The double-bouligand designs are benchmarked against bouligand, conventional rectilinear counterparts and monolithic cast. The hypothesis that the spatial arrangement of the proposed complicated double-Bouligand architecture of concrete materials can enable favorable mechanical properties is impressively confirmed by the tests performed and the methods used.

The work is of extraordinary relevance to structural design and architecture. It is an independent and highly innovative contribution to advance the adoption of AMC and to inspire engineers and architects for designs with a novel "logic of form" based on material and resource efficiency. Data analysis, results and conclusions are valid. The chosen methods are convincing and described in detail, the reproducibility is given. The supplementary information is very good and enriches the main work.

a

I recommend the acceptance of the paper.

23/07/09

Harald Kloft

Response. Thank you for your thorough understanding of the paper and the elaboration of its implications. We appreciate the contextualization provided for the community as to what additive construction means in the place and times of our lives in further developing our urban habitat and the built environment. We are also thankful for the recognition of the novelty of the proposed three-dimensional deposition patterns inspired by nature and the opportunity presented in these non-homogeneous systems, where weak layer bonding of the materials takes place in the additive process.

Indeed, the design and implementation of the material, the extrusion process, and the software package that enable novel designs of construction materials, or simply the technology, have enabled the adoption of more natural design principles at the micro scale.

We are grateful for the major understanding that the novelty of the approach accompanied by testing of the constructed hypothesis here (in this case using double-bouligand architecture), is a way to establish directions for new possibilities at a large scale, and indeed why this effort was taken at the largest laboratory scale possible and not at a smaller desktop or benchtop robot scale. Most certainly, we are thankful for better redefining our work as novel 'logic of form', which we have been alluding to in the manuscript as 'purposeful design' for the proposed approach. We envision that the current study, although limited compared to the future research in the field in the next decades, to help expand the opportunities in the design and automation domain for civil and urban infrastructure. The designs can, in principle, be expanded in form and to alternative materials and processes at large scales.

Reviewer #3 (Remarks to the Author):

The paper describes a study to determine the effect of an architected material composition achieved by 3d printing in bouligand and double bouligand architectures on tensile strength and fracture toughness of cementitious composites, with the aim of increasing particularly the latter and thereby overcoming a significant drawback of the use of these materials in building structures. The approach is inspired by architected materials found in nature.

Response. We appreciate this understanding of the paper and the approach.

Although this concept is innovative in its consideration of material properties and use of 3d printing methods to achieve them, I have to advise against publication of the manuscript in its current form, for several reasons.

First of all, the paper is based on very little data. Unfortunately, it is not explicitly stated how many specimens were tested. The supplementary data just mention 'minimum of 2 per specimen type'. In my opinion that is too little to warrant any solid conclusion or perform any statistical analysis, it can merely point into a direction. Considering the procedure is not overly complicated, a considerably higher number of experimental results would be expected for a publication in this journal.

Response. We appreciate this comment and, as advised, have performed extensive additional experiments. We have obtained a higher number of data points for both Bouligand and Double-Bouligand in terms of both fracture toughness (using SENB with 6 samples each) and strength (using 3PB with 4 samples each). We have added this information in the Main **Line 418 – 422**, using statistical analysis incorporating the additional data set. **Fig. 4** is entirely updated along with the revisions in the **Main Lines 140 – 147, 160 – 161, 195 – 196, 200, 215 - 222**. The **'significance' among the various designs was** shown using a star-bar (*) inside **Fig. 4b, 4e, 4f**.

Furthermore, besides the destructive testing results, there is very little reporting on supplementary measurements, properties, etc. There is no data on the fresh or hardened state properties of the printing mixture,

Response. Thank you for making these suggestions about supplementary reporting of the properties. We have conducted additional experiments on both fresh and hardened properties as follows and revised the manuscript accordingly:

- **Fresh 3DP properties:** A Cone Penetration test ^[12,13] one of the common fresh property tests in concrete additive manufacturing, was conducted to measure the yield stress properties of the materials immediately after extrusion. To do this, a rectangular geometry was designed and 3D-printed and was tested using a fabricated cone under a mechanical testing unit using a 0.5 mm/s displacement rate. The test has been previously established as suitable to examine the yield stresses ranging from 100 Pa to 100 KPa.^[14] A yield stress of 20.2 ± 1.0 KPa was found for the accelerated concrete mixtures used in this study. The method and results are discussed in **Main Line 424 – 425** as well as in a new **Supplementary Fig. 8b** and **Lines 376 – 387** in the **Supplementary Information**.
- Furthermore, isothermal calorimetry was additionally performed and added to study the hydration of the same extruded concrete in the 2-K system. See **Main Line 425 – 426** and **Supplementary Fig. 8a & Line 369 – 375**.
- **Hardened 3DP properties:** 7-day compressive strength property was conducted and assessed on the same extruded material used in this study using a 3"x 6" cylindrical sample tested at 0.6 mm/min. A compressive strength of 52.73 ± 1.62 MPa and Young's modulus of 25.59 ± 1.72 GPa were found. The methods and results are discussed in **Main Line 426 – 429**, a new **Supplementary Fig. 8d,e**, and **Lines 392 - 399** in **Supplementary Information**.

no data on interface or bulk material strength, material density or object density (only some very limited data in the supplementary notes file), etc.

Response. We have conducted additional experiments as follows and incorporated them in the revised Supplementary Information for clarity and comparison with the rest of the materials used in the literature:

- **Bulk material strength:** As discussed in previous comments, 7-days bulk compressive was conducted and is added (See the new Supplementary **Figure 8d, e** and **Lines 397 – 398**). In terms of the bulk material tensile strength and fracture toughness properties, those were previously reported for 'cast' in Fig. 4b and 4e.
- **Material density:** We conducted new experiments and have reported the material's fresh and hardened density in the new **Supplementary Fig. 8c** and **Lines 388 - 389**.
- **Object density:** We have reported the object density for various designs in the supplementary information in the text. Please see **Supplementary Information Lines 389 – 390**. Nevertheless, the total porosity in the 3D-printed samples was previously reported in Supplementary Fig. 4, corresponding to object density.

¹² Ivanova, I., *et al.*, 2022. *Cement and Concrete Research*, 156, p.106764.

¹³ Lootens, D., *et al.*, 2009. *Cement and Concrete Research*, 39(5), pp.401-408.

¹⁴ Reiter, L., *et al.*, 2022. *Cement and Concrete Research*, 157, p.106802.

- **Interface properties:** The properties obtained from perpendicular lamellar (\perp) samples are the properties of the interface, previously reported as strength and fracture toughness (Fig. 4b and 4e) in the original manuscript.

Some hardened state properties used in analyses are apparently based on 28-day testing, while the specimens themselves were tested at 7 days. Considering the use of accelerators, it is not evident they can be accurately calculated from each other. The effect of printing vs casting for these properties is not discussed. And the actual numbers on these properties are not given.

Response. Thank you for this suggestion. We have conducted new experiments for 7-day compressive strength and Young's modulus (E). We have used the updated 7-day E value for the calculations accordingly and have revised Fig. 4 and all the findings. More specifically:

The use of 7-day compressive properties directly given the use of accelerator: We have used the newly conducted 7-day experiments to directly calculate the elastic modulus used in Eq. 4 of the Supplementary Information as suggested. This has led to an E value of 25.59 ± 1.72 GPa. The Supplementary Information **Lines 336 - 337** are revised accordingly.

Accounting for difference in porosity: The difference in porosity is small and is reported in the revised manuscript (Please refer to Supplementary **Lines 389 - 390**). The effect of casting vs. printing is reported by comparing the 3D-printed samples with cast benchmarks and presenting the raw data in Fig. 4 (not normalized with density).

The printing quality of the specimens would generally be considered sub-standard due to the large air voids (visible in the pictures, as well as in the supplementary data). This raises the question how applicable the results would be in general, as most printing operators would not print this way for a variety of reasons.

Response. In other words, the intention of this manuscript was not to replicate what industry does (nevertheless, the field does have a clear picture about the degree to which the printing induced flaws take place in the field structures) rather to develop flaw-tolerant materials in the sense that using certain design schemes, the properties can remain insensitive to a certain degree of defects as shown in Fig. 4.

To elaborate, we would like to iterate a few points:

- More importantly, this manuscript makes the case that 'weak' interfaces can be harnessed to improve mechanical properties, while many other works in the literature aim to enhance the interfacial bond strength.^[15,16] This point has been previously discussed in the introduction Lines 73 - 77. Thus, we did not intentionally aim to eliminate the porosity (i.e., preventing artificially enhancing the interfacial properties by manipulating/perfecting the print quality) that often naturally occurs in the printing process in the lab and the field. In other words, here, we don't attempt to create perfectly dense systems so as not to overestimate or overly rely on 'perfect' interfacial properties that are also not realistic in application. Here, we intended to harness the role of the weak interface and to understand (if/as they happen) how they can be used to enhance

¹⁵ Yu, K., et al., 2021. *Cement and Concrete Research*, 143, p.106388.

¹⁶ Cao, X., et al., 2022. *Automation in Construction*, 141, p.104392.

the material's mechanics from a research perspective. We have already briefly discussed these in the Main discussion of Fig. 1, as well as the conclusions.

- We understand this comment and that the presence of void is generally not desired in engineered material, including in additively manufactured concrete. Nevertheless, they can occur in the field as little progress has been made in the non-destructive testing of real-world 3D-printed structures in real time. Indeed, this is one of the main issues in the field of concrete AM, as the reviewer touched on. Despite the current efforts in developing guidelines, code, and standardized property (e.g., at ACI, ASTM, and RILEM), there currently remains the lack of unified standard specifications for defining an 'optimal/sufficient' property (e.g., density, porosity, and strength in comparison to the 'acceptable' range known for cast concrete). This is, in part, due to the versatility of the processes, materials, and types of flaws and defects that arise from a layer-wise manufacturing process. This is clear from the current reliance on mechanical testing in the field at the moment. On the other hand, posing 'optimal' ranges can limit the creative nature of engineering of additively manufactured materials and hinder novel design opportunities, such as those with the notion to harness internal flaws as a means to improve material properties such as those in natural counterparts.
- Moreover, the body of work in the literature emphasizes the presence of large porosity in laboratory conditions ^[15,16], whereas there is little study of the microstructure of printed structure in published papers, given those structures are not often the subject of academic/laboratory research or the data remains private, unfortunately, due to often litigious reasons.
- Nevertheless, the reported porosity in this work is on par with several other works for robotic additive manufacturing of concrete materials by researchers across several scales. It has been shown that the presence of voids (~ 5 – 13 %, calculated from micro-CT analysis) is common, given the current state of control in extrusion-based concrete AM.^[17,18,19,20] Our results represent a range ($7.7 \pm 0.08\%$), as shown in this figure, on par with 'commonplace/mainstream' additively manufactured concrete. We believe this is a good comparison with the rest of the current literature. We have added the brief discussion in Supplementary Information Line 220 – 222 to better contextualize this matter based on this comment.

The printing method also results in specimens of different geometries. Although roughly a solid beam, the irregular surfaces may influence the effective sizes used in calculations as well as the local stress distributions. It is also not clearly indicated where the notches are applied respective to the interfaces, and why.

Response. Thanks for the opportunity to clarify.

- Effect of geometry on effective size: The printing can always result in a slight difference in cross-sectional area compared to prismatic cast beams. To account for this, the cross-section of printed samples was carefully measured to account for the actual size,

¹⁷ van den Heever, M. *et al.*, 2022. *Cement and Concrete Research*, 153, p.106695.

¹⁸ Kruger, J., *et al.*, 2021. *Additive Manufacturing*, 37, p.101740.

¹⁹ Lee, H., *et al.*, 2019. *Construction and Building Materials*, 226, pp.712-720.

²⁰ Mohan, M.K., *et al.*, 2023. *Cement and Concrete Composites*, 140, p.105104.

not 'an equivalent rectangular size', by taking sufficient measurements across the width and height. We appreciate the opportunity to clarify.

- Effect of geometry on stress distribution and notch position: The fracture is driven by the critical stress intensity factor at the notch length (sharp notch)^[21]. The sharp notch was carefully made and was placed such that the tip resides at the end of the first 'sacrificial' of the first layer (between the first and 2nd layer) – residing a continuous shape tip at both the core and the edge.
- Regarding the notch location relative to the interface: In the double-Bouligand, Bouligand, and perpendicular cases, the notch was introduced at the interfaces between two filaments, and in the parallel case, the notch was introduced perpendicular to the interfaces and filaments. We have added this discussion in the **Supplementary Line 307 – 314**.

The experimental procedure can also be questioned. A 3-point bending set-up is not suitable to compare and analyse post-fracture behaviour if the fracture does not remain in the area under the loading point – which it does not for the (double-)bouligand architectures – because the stress levels reduce fairly quickly when moving away from the loading point. A set-up featuring larger areas of constant tensile stress would have been more suitable (4p bending or ideally uni-axial tension).

Response. Thanks for the opportunity to clarify. It is pointed out that “A 3-point bending set-up is not suitable to compare and analyse post-fracture behaviour”. The 3PB test was *not* used to analyze the fracture or to report fracture toughness. It was used to report strength (modulus of rupture). To analyze the fracture, the single-edge-notch-bent (SENB) test was used.

We disagree that a tensile test is more suitable. The SENB test is the most commonly used test for examining fracture and has been used for a long time (several decades). It has also been used in characterizing the fracture toughness of cementitious materials,^[22,23] in numerous studies in which the crack is indeed subjected to deflection as it interacts with the microstructure.^[24,25,26] For instance, tortuous cracks have been observed in SENB testing of concrete due to crack deflection along weak surfaces around aggregate, aggregate interlocking, and bridging.^[25,26] Thus, the presence of crack deflection does not disqualify the test or the use of it. Furthermore, in SENB of fiber-reinforced concrete (FRC), the interaction of crack with fibers in addition to aggregates makes the crack more tortuous than in plain concrete.^[24] For these reasons, we have opted for a SENB test as opposed to a direct tension test. The use of this test, indeed, allows us to compare our properties with other types of concrete in the literature, as conducted in Fig. 5.

More specifically, several works have been published where crack deflection was observed in SENB testing of architected materials.^[27,28] In such scenarios, the crack remains locally in Mode I, though it can globally be in Mode II and non-planar (as discussed in Supplementary

²¹ Dunn, M.L., *et al.*, 1997. *International Journal of Solids and Structures*, 34(29), pp.3873-3883.

²² Shah, S.P., 1990. *Materials and Structures*, 23, pp.457-460.

²³ Hoover, C.G. and Bazant, Z.P., 2013. *Engineering Fracture Mechanics*, 110, pp.281-289.

²⁴ Ren, H., *et al.*, 2023. *Construction and Building Materials*, 409, p.134053.

²⁵ Shah, S.P. and Ouyang, C., 1994. *Annual review of materials science*, 24(1), pp.293-320.

²⁶ Nikbin, I.M., *et al.*, 2014. *Construction and Building Materials*, 52, pp.137-145.

²⁷ Munch, E., *et al.*, 2008. *Tough, bio-inspired hybrid materials*. *Science*, 322(5907), pp.1516-1520.

²⁸ Suksangpanya, N., *et al.*, 2018. *International Journal of Solids and Structures*, 150, pp.83-106.

Figure 3 about the contributions of such deflection or twisting).^[29] These points have previously been discussed in the original manuscript. However, we will be glad to discuss this point further if necessary.

The applied displacement-controlled loading rate is very low, favouring crack path diversions. However, at higher (more common) rates the observed differences may disappear. At the very least, this should be discussed, and preferably also studied experimentally. The relevance of increasing the fracture toughness of (printed) concrete this way is questionable.

Response. Thanks for the suggestion to discuss the effect of the loading rate. We have added two sentences about the choice and effect of the loading rate, elaborating on the effect of the loading rate on capturing the cracking phenomena in **Supplementary Information Lines 315 - 317.**

It is correct that at high loading rates, crack propagation can be very fast and the information about the fracture process can be lost. For these reasons, many studies on fracture mechanics of brittle and quasi-brittle materials (unless they deal with dynamic fracture), work with static and quasi-static loading at relatively low rates. The typical loading rates used in the literature for concrete are in the order of **0.024 to 0.1 mm/min** for the sample depth varying from 40 mm to 200 mm.^[30,31,32,33,34,35] Our rate is within this range (**0.05 mm/min**) with a sample depth of 195 mm.

To contextualize the matter further, the loading rates of brittle and quasi-brittle cementitious materials have been considered as *fast* (around 1 - 2 sec), *usual* (around 6 – 10 min), *slow* (around 3 – 5 hr.), and *very slow* (around 2 – 3 days) based the ‘time-to-peak’ criterion established by Prof. Bazant.^[36] The time-to-peak used in the fracture test here (with 0.05 mm/min loading rate) yields 226 – 849 sec (592 ± 156 sec), which is in the range of ‘usual’ with the 360 – 595 sec.^[36] This should be a reasonable rate as the majority of the laboratory experiments for cementitious materials are commonly conducted for this time-to-peak range of 5 – 10 min.^[36,37]

Although we have not used ‘*slow or very slow*’ rates here, it is not uncommon in cementitious material fracture research to use them, especially in brittle materials.^[38] Those are done to capture a ‘snap-back’ behavior rate as low as 0.6 micron/min, which may have to be used and provide meaningful information about the nature of crack propagation and the fracture energy.^[39]

On the other hand, ‘*fast*’ loading rates not only lead to a loss of information about fracture processes, as mentioned earlier, but they are also meaningful for applications where higher loading rates or dynamic conditions are present or being studied. In other words, from a research perspective, fast loading rates cannot be very useful in studying brittle or quasi-brittle failure unless the applications determine the research.

²⁹ García-Álvarez, et al., 2012. *Sadhana*, 37, pp.187-205.

³⁰ Ruan, Y., et al., 2018. *Construction and Building Materials*, 162, pp.663-673.

³¹ Xie, J., et al., 2022. *Construction and Building Materials*, 323, p.126612.

³² Yin, Y., et al., 2019. *Engineering Fracture Mechanics*, 211, pp.371-381.

³³ Qing, L., et al., 2022. *Fatigue & Fracture of Engineering Materials & Structures*, 45(2), pp.400-410.

³⁴ Zhang, J., et al., 2023. *Construction and Building Materials*, 401, p.132699.

³⁵ Sakai, M. and Ichikawa, H., 1992. *International journal of fracture*, 55, pp.65-79.

³⁶ Bazant, Z.P. and Gettu, R., 1992. *ACI Materials Journal*, 89(5), pp.456-468.

³⁷ Hoover, et al., 2013. *Engineering fracture mechanics*, 114, pp.92-103.

³⁸ Verma, R.K. et al., 2021. *International Journal of Rock Mechanics and Mining Sciences*, 147, p.104897.

³⁹ Verma, R.K. et al., 2021. *In ARMA US Rock Mechanics/Geomechanics Symposium* (pp. ARMA-2021). ARMA.

The relevance of increasing the fracture toughness of (printed) concrete this way is questionable. It seems to come at the cost of tensile strength and stiffness. Particularly the latter is a very important parameter for structural concrete engineering.

Response. Thank you for the opportunity to clarify.

- We are not increasing fracture toughness by the way and rate we test the beams. As discussed above, the test is the most common, and the rates are the most usual. We chose the appropriate and established rate and compared the fracture toughness of the proposed design with benchmark cast and lamellar cases, all of which tested at the same rate. To further validate the results, Fig. 5 was proposed for comparison of the results with the commonly reported properties of conventional cementitious materials. Thus, all data, including the proposed and benchmarked cast, were then compared with the literature data in Fig. 5 to compare the benchmark cast. We must note that our cast case, to which the proposed designs of the same material are being compared, is the best of the literature, not the worst in terms of fracture toughness. The improvements presented in Fig. 5 are also shown as the absolute fracture toughness values (and not normalized with respect to density) for transparency.
- Broadly speaking, in terms of trade-offs between the fracture toughness and stiffness or fracture toughness and strength in engineering materials, both properties are often improved at the same time. We further investigated the effect of proposed designs on stiffness based on the 3PB test, and found that the stiffness is not sacrificed at the cost of improving fracture toughness. **Figure 1R** below presents the data on stiffness obtained from the slope of the load-displacement plot, demonstrating that the double-bouligand, bouligand, and parallel cases present a statistically similar stiffness compared to the cast.

Fig. 1R. Comparison of stiffness of proposed designs vs. cast

The tensile is generally considered less important as we cannot rely on the tensile strength of concrete anyway. In practice we should always consider that the concrete may be fractured due to other causes such as shrinkage, creep, and (related) forced deformations.

Response. Thank you for the opportunity to clarify. It is absolutely correct that we currently don't rely on concrete tensile *strength* properties in the design of concrete structures and rely

mostly on compressive properties. This long-standing weakness of the material has in a way determined the fate of the design of concrete structures. However, cracking, indeed, is most common in cementitious material in mode-I (opening) in various tensile or flexural loading conditions. The entire point of this study is to demonstrate the utility of additive manufacturing to improve resistance to cracking that takes place in mode-I, thus presenting the notion to harness flaws in improving the weak crack-resistance characteristics in these materials; it is also true that this long-standing weakness of the material has determined the fate of the design of concrete structures.

We also appreciate the point that cracking, and fracture can take place in concrete from a wide range of intrinsic conditions (e.g., creep, shrinkage) at early and later stages. These intrinsic cracking potentials of concrete only further highlight the utility and implication of this study in which we examine how to exploit the design to enhance the resistance to crack (propagation) once there is a pre-existing notch (crack – whether it is initiated by a notch or by shrinkage within the material). This study very much focuses on propagation resistance (using SENB in mode I) and is indeed applicable to conditions where constraints can induce early-age deformation (e.g., from adhering to the substrate or fast evaporation rates in exposed additively manufactured concrete).

We have added a brief statement highlighting this context in Main **Lines 339 – 340 and 344 - 356**. Thank you for this suggestion.

But also the relevance of fracture toughness should for structural concrete should not be overestimated. In structural engineering we typically consider loads to be (quasi) static, thus the energy dissipation in fracture becomes less important: something either breaks or not. Fracture toughness would only be relevant in dynamic or (short duration) temporary loadings such as explosions or earthquakes. But even then, it can be debated (and should be analysed in such a paper) whether the obtained increase in fracture toughness is relevant for such cases. Because although the authors claim the developed material design methodology can obviate the need for fibers or reinforcement (bars), I expect the energy absorption capacity of such additions still to be many times higher than the fracture toughness found in these architected materials.

Response. Thank you for the opportunity to clarify.

- Relevance of fracture toughness/mechanics. It is true that the current structural engineering design code does not consider fracture mechanics except for a few instances. We would like to iterate that this manuscript does not propose a revision to the design code to consider fracture toughness. We are studying the effect of design and internal flaws in the process of fracture, which, involves energy dissipation, load-bearing capacity upon formation of a crack, and how to induce a non-catastrophic and non-brittle response in unreinforced concrete materials. Fracture toughness and strength of concrete are both size-dependent (known as type 2 and type 1 size effects) ^[40,41,42] and thus determine the load at which 'whether something breaks or not,' thus, should not be entirely discounted in the design of concrete structures at small or large scales. Thus, we disagree with the lessening of fracture toughness and its importance at either the material/component level (this study) or at the structural level (future studies). Fracture toughness is always going to be relevant in studying the failure of structures at any scale

⁴⁰ Bažant, Z.P., 2000. Size effect. *International Journal of Solids and Structures*, 37(1-2), pp.69-80.

⁴¹ Bazant, Z.P. and Planas, J., 2019. *Fracture and size effect in concrete and other quasibrittle materials*. Routledge.

⁴² Bažant, Z.P., et al., 2021. *Quasibrittle fracture mechanics and size effect: A first course*. Oxford University Press.

or loading conditions and is not limited to dynamic conditions. Everything that fails eventually becomes the subject of study in fracture mechanics.

- By studying fracture processes in 3D-printed concrete (a subject that is significantly under-studied), we are not overestimating the relevance of fracture toughness in structural concrete. We are highlighting that studying fracture characteristics and resistance curves can be far more useful to understand how these materials and structures are going to fail and how to improve upon them given the presence of interface, in comparison to overly emphasized/reported compressive strength that is highly insufficient to improve our understanding of failure and damage. We are simply highlighting the relevance of fracture phenomena in 3D-printed concrete as a new way to consider concrete that can dissipate additional energy in the process of fracture under tension (mode I). This is a new approach that has not been taken before in developing concrete that can respond better to fracture – not just to crack initiation but to crack propagation (e.g., through preservation of stable crack growth, thus preventing progressive damage or abrupt failure.)
- In conventional concrete, cracked concrete sections are not considered in the design of reinforced concrete, due to this very lack of resistance to fracture. This study scratches the surface that the use of additive manufacturing and purposeful designs of architected materials can provide new pathways to enhance resistance to cracking and fracture toughness. There are some examples of a great body of literature where resistance to cracking is important in concrete (in the sense that it does not just fail or not, and the ability to take load besides initial crack is of great importance and is the subject of study). For instance, ultra-high-performance concrete (UHPC) is used in flexural loading conditions in civil infrastructure, where our community has particularly studied fracture phenomena and fracture toughness as well as ‘residual’ (tensile) load-bearing capacities at the material/component. ^[43,44,45,46]
- The need for fiber/reinforcement. Nowhere in the manuscript, have we indicated that the proposed architected materials remove the need for fibers or outperform fiber-reinforced concrete. The statement, in conclusion, highlights not necessarily requiring fiber to achieve a non-brittle response: “*This fosters an approach towards the design of non-brittle construction material without requiring the addition of fibers or reinforcement*”. In other words, these designs enabled by robotic additive manufacturing can provide additional pathways to achieve improved properties. They are not mutually exclusive with the use of fibers.

We have added a brief statement highlighting this context in Main **Lines 337 - 338**. Thank you for this suggestion.

Owing to the small number of tested samples and tests performed, not much can really be concluded from the work. This is perhaps the reason why several textual formulations appear repeatedly (basically the conclusion that a (double-)bouligand architected material structure improves the fracture toughness).

⁴³ Zhang, Y., et al., 2021. *Engineering Failure Analysis*, 120, p.105076.

⁴⁴ Honarvar Gheitanbaf, E., 2011. *University of New Mexico*.

⁴⁵ Yang, S., et al., 2023. *Cement and Concrete Composites*, 136, p.104860.

⁴⁶ Ahmad, S., et al., 2024. *Construction and Building Materials*, 417, p.135327.

Response. Additional experiments on both fracture and strength were conducted. The new experiment support the previous claims that double-bouligand design help improve the fracture toughness of concrete compared to cast and the literature. Thus, the key claims of improved fracture toughness in the initial submission remains. Please see the first comment and corresponding revisions. Please see Main Lines 195 - 196 and revised Fig. 4.

The paper, finally, would need some restructuring, as the method comes at the end (should be before results), the method is described much to generally, and there are no conclusions (actually the discussion seems to hold the conclusions, but this leaves for a very brief discussion that could have gone more in depth).

Response. We believe the paper is structured per Nature Communication Guidelines in which the methods appear at last, and the conclusions are discussed at the last paragraph. This is the common format in this journal.

The method is described in the Main, and more specifics are given in the Supplementary Information to maintain brevity in the Main document. The discussion is followed by the conclusion. This is a common thread in the narrative of the papers published in this journal.

The discussion can be extended; however, again, for brevity (and avoiding repetitions that was mentioned in the earlier comment), the key points are discussed in the Main discussion, and the relevant figures in the Supporting Information (for instance, Supp. Fig. S3). This is to maintain the word limit in the Main to the key findings of the paper.

Please refer to the uploaded reviewer files for further detailed comments.

Response: We have read, revised, and addressed all the comments in the two PDF of the Main and Supplementary. They are addressed and a brief elaboration is provided in the summary below.

Summary:

- Additional details of methods are added. See Main Lines 416, 418 – 422 & 424 – 429 and Supplementary Lines 307 – 313, 315 – 317, 321 – 322, & 369 - 399.
- Conclusion section is added. See Main Line 308. All other corrections and edits are highlighted.
- Other comments that were not addressed in the earlier response (cited based on the pdf comments and original Lines) :

Supplementary Materials:

- Line 47 – RAPID vs. other codes: The programming language could be different than RAPID for robots, other than ABB. (See Supplementary Line 26)
-
- Line 47 – Use of 2-K system: The helical architected materials ‘could’ be fabricated with a 1-K system but the quality would not have been the same based on our experience given our efforts using the same materials in the 1-K system due to lack of acceleration. Thus, we chose to work with 2-K system. In addition,

the columns and non-planar structures demonstrated in this study would have been impossible to fabricate using the same material with the 1-K system.

- Lines 114 and 37 – Point of Supplementary Note 2 and 1-K system discussion: We believe the broad audience can benefit from the baseline comparison and the motivation for the use of the 2-K system.
- Lines 143 and 123 – Effect of the accelerator on long-term strength and durability and comparison of 2-K vs. 1-K: The relevance of comparison is the use of the 2-K system allows for tailored morphology for achieving more geometrically complex architected materials that benefit from early-age shape stability due to acceleration. The comparison of the two systems does not disregard or make statements about the effects of long-term durability. While 1-K systems have been used for full-scale buildings, they still have limited build-up rates compared to the 2-K systems. Thus, the statement in line 123 remains valid (Line 124 in updated Supplementary).
- Line 287 – 288 – Suggestion to move a sentence to method in the main paper. Based on your suggestion, the sentence “The beams were then covered ... curing until tested” is moved to the method section of Main Line 394 – 396.
- Line 156 – How was G_c/G_c^m calculated: We have added the reference that includes the equations used in this study to Fig. S3 caption. (Supplementary Line 175)
- Line 313 – Past the peak load: is the area past the softening point; thus, A_{pl} will not be zero.

Main:

- Line 38 – Mutual exclusivity of properties: Please see the earlier comments on toughness vs. strength vs. stiffness. The statement about mutual exclusive is with regard to challenges in improving one property without significantly reducing another. The statement is revised (Line 39) to be broader for the introduction. As for steel, again, the comparison is not between the material but in improving the material properties. In the case of steel, plasticity allows for high toughness that is different than fracture toughness.
- Line 70 – Schematic response: The word ‘schematic’ was used to avoid this confusion. We have further revised the caption to clarify this is not the actual response.
- Line 97 – Figure 2,c,d,g,h: These are self-created original images so no citations are needed. Figures 2a,b,e,f are open-source images.
- Line 109 – 2-K system uniqueness: We have not claimed our 2-K system is exactly unique and are simply explaining the uniqueness of the process vs. 1-K.
- Line 224 – Crack tortuosity: We have made observations of the fracture plane. It’s a common phenomenon to report the observation of ‘roughness’ of the fracture surface.
- Lines 229, 321, 323 – Brittle failure: We disagree that the two responses in Fig. 4 are brittle in the same way. We have revised the manuscript to reflect the suggestion to ‘less brittle’. Some of the repeated information was removed.

- Line 298 – Ashby plot references: References to the works used to compare to other cementitious materials are added and reported in the caption (Main Line 304 – 306).
- Line 330 – fibers: See earlier comments. The sentence is further revised to better contextualize the approach for the use of architected materials is not orthogonal to the use of fibers.
- Line 359 – Pitch angle choice: This was based on a preliminary study that informed of such an angle at a smaller scale. The inclusion of those preliminary results (not published) guided this current research.
- Line 369 – Materials composition: Please see the earlier comments on the addition of fresh rheological properties and hydration rates.
- Line 390 – The precise number of samples is reported for each type of design as requested. We also added this number per type of test (fracture vs. strength) in Methods.

REVIEWERS' COMMENTS

Reviewer #1 (Remarks to the Author):

The paper is significantly improved, all comments are tackled adequately and can be accepted in the current version. Very inspiring work, indeed!

Reviewer #3 (Remarks to the Author):

-The authors have extensively addressed my comments to the previous version of the manuscript, as well as those of the other reviewers. They have provided additional experimental research and additional details on methods and theory. They have furthermore provided clarifications were needed. Together, this has significantly improved the quality of both the main article and the supporting information. The manuscript can therefore be accepted for publication.

However, please check the figure numbering (there are 2 Figures '4').

Response to Reviewer's Comments:

Thank you for the thorough consideration and reviews. We have made additional changes to the manuscript as requested per Nature Communications checklist and editorial policy. All new changes are made in cyan.

REVIEWERS' COMMENTS

Reviewer #1 (Remarks to the Author):

The paper is significantly improved, all comments are tackled adequately and can be accepted in the current version. Very inspiring work, indeed!

Response: Thank you. We appreciate the input and the thorough review provided for this manuscript. They have helped improve the quality of this study.

Reviewer #3 (Remarks to the Author):

-The authors have extensively addressed my comments to the previous version of the manuscript, as well as those of the other reviewers. They have provided additional experimental research and additional details on methods and theory. They have furthermore provided clarifications were needed. Together, this has significantly improved the quality of both the main article and the supporting information. The manuscript can therefore be accepted for publication.

Response: We are thankful for the detailed comments, suggestions for improvements, and the opportunity to make clarifications. We appreciate the time and effort invested in reviewing this manuscript.

However, please check the figure numbering (there are 2 Figures '4').

Response: Thank you. We have revised accordingly.